# Incidence patterns of orofacial clefts in purebred dogs

**Nicholas Roman[1¤], Patrick C. Carney[2☉], Nadine Fiani[2], Santiago Peralta [2☉]***

**1** College of Veterinary Medicine, Cornell University, Ithaca, NY, United States of America, **2** Department of Clinical Sciences, College of Veterinary Medicine, Cornell University, Ithaca, NY, United States of America

☉ These authors contributed equally to this work.
¤ Current address: Cheshire Cat Hospital, Broomfield, CO, United States of America.
* sp888@cornell.edu

**Data Availability Statement:** All relevant data are within the paper and its Supporting Information files.

**Funding:** The author(s) received no specific funding for this work.

## Abstract

Cleft lip (CL), cleft palate (CP) and cleft lip and palate (CLP) are the most common types of orofacial clefts in dogs. Orofacial clefts in dogs are clinically relevant because of the associated morbidity and high newborn mortality rate and are of interest as comparative models of disease. However, the incidence of CL, CP and CLP has not been investigated in purebred dogs, and the financial impact on breeders is unknown. The aims of this study were to document the incidence patterns of CL, CP and CLP in different breeds of dogs, determine whether defect phenotype is associated with skull type, genetic cluster and geographic location, and estimate the financial impact in breeding programs in the United States by means of an anonymous online survey. A total of 228 orofacial clefts were reported among 7,429 puppies whelped in the 12 preceding months. Breeds in the mastiff/terrier genetic cluster and brachycephalic breeds were predisposed to orofacial clefts. Certain breeds in the ancient genetic cluster were at increased odds of orofacial clefts. Male purebred dogs were at increased odds of CPs. Results confirm that brachycephalic breeds are overrepresented among cases of orofacial clefts. Furthermore, geographic region appeared to be a relevant risk factor and orofacial clefts represented a considerable financial loss to breeders. Improved understanding of the epidemiology of orofacial clefts (frequency, causes, predictors and risk factors) may help in identifying ways to minimize their occurrence. Information gained may potentially help veterinarians and researchers to diagnose, treat and prevent orofacial clefts.

## Introduction

Orofacial clefts are abnormal fissures of oral or facial structures that occur due to incomplete fusion of tissues during embryonic development and have been described in multiple mammalian species, including dogs. Orofacial clefts in dogs are of major clinical relevance because of the associated morbidity and high newborn mortality rate due to aspiration pneumonia, failure to thrive or euthanasia [1–5]. Orofacial clefts in dogs are also of comparative and translational research interest as they represent useful models of analogous disease in humans [6–8].

**Competing interests:** The authors have declared that no competing interests exist.

Orofacial clefts in dogs are recognized shortly after birth based on signs observed by the breeder or clinician (e.g., drainage of milk from the nares during or after nursing; gagging, coughing or sneezing while eating) and visual examination [2, 9]. Surviving animals usually require nutritional support and palliative care until surgical repair can be performed at a later age [10, 11]. Cleft repair is technically complex, often requires multiple surgeries and is usually exclusive to specialized referral centers [10, 11]. Therefore, the loss of animals due to medical complications or euthanasia (or the associated cost of nursing and surgical repair) represents a potentially substantial financial burden to breeders.

The most commonly described orofacial clefts in dogs involve the upper lip, incisive bone and midline of the hard and soft palate [2, 12–14], although other types of defects have been sporadically reported [15–17] (Fig 1). Cleft lip (CL) is the clinical term used to describe defects that affect the primary palate (i.e., lip and alveolar process), whereas cleft palate (CP) refers to midline defects present in the secondary palate (i.e., hard and soft palate) [5, 12]. The term cleft lip and palate (CLP) applies to defects that simultaneously affect the primary and secondary palate [5, 12].

As in humans, the etiopathogenesis of orofacial clefts in dogs is usually regarded as complex, involving possible interactions between genetic and environmental mechanisms [2, 18, 19]. In general, orofacial clefts are phenotypically and genetically diverse and often show incomplete penetrance [9, 15, 20]. With few exceptions [4, 6, 21–23], the genetic mechanisms of orofacial clefts in dogs remain unexplored. It has been suggested that defect phenotype (i.e., CL, CP or CLP) is associated with skull type in dogs (i.e., brachycephalic, mesaticephalic or dolichocephalic) [12]. Other possible variables that could be associated with phenotype include sex, breed, and genetic cluster [24, 25]. However, these possible associations have not been investigated in dogs.

Environmental factors that have been implicated with orofacial clefts in dogs include nutritional (e.g., hypervitaminosis A, folic acid deficiency), drugs (e.g., corticosteroids), intrauterine mechanical trauma, toxins and viruses [9, 15, 26, 27]. Some of these factors may be related to husbandry practices (i.e., nutritional, trauma, drugs), but others might be associated with geographic location (i.e., toxin and virus exposure).

Despite their clinical and comparative relevance, the incidence patterns of CL, CP and CLP have not been investigated in purebred dogs, and the financial impact on breeding programs is unknown. Understanding the incidence patterns of orofacial clefts in purebred dogs may help better elucidate the possible etiologic mechanisms involved, assist in the development of prevention strategies, and help minimize the financial burden on breeders. Therefore, the aims of this study were to determine whether the incidence of CL, CP and CLP in purebred dogs is associated with sex, skull type, genetic cluster, and geographic location; and to estimate the financial impact of orofacial clefts in breeding programs in the United States. The hypothesis was that incidence of orofacial clefts would be higher among dogs belonging to brachycephalic breeds regardless of sex.

## Materials and methods

### Data collection

A survey questionnaire was developed using a university-specific implementation of a commercially available online survey tool with demonstrated reliability with most browser and Internet access providers (Qualtrics Web Survey Tool, Qualtrics, Provo, UT). The questionnaire (S1 File) was reviewed by Cornell University's Human Research Protection Program and was found not to meet the definition of human participant research and was therefore not subject to review or oversight by the Institutional Review Board for Human Participants. The

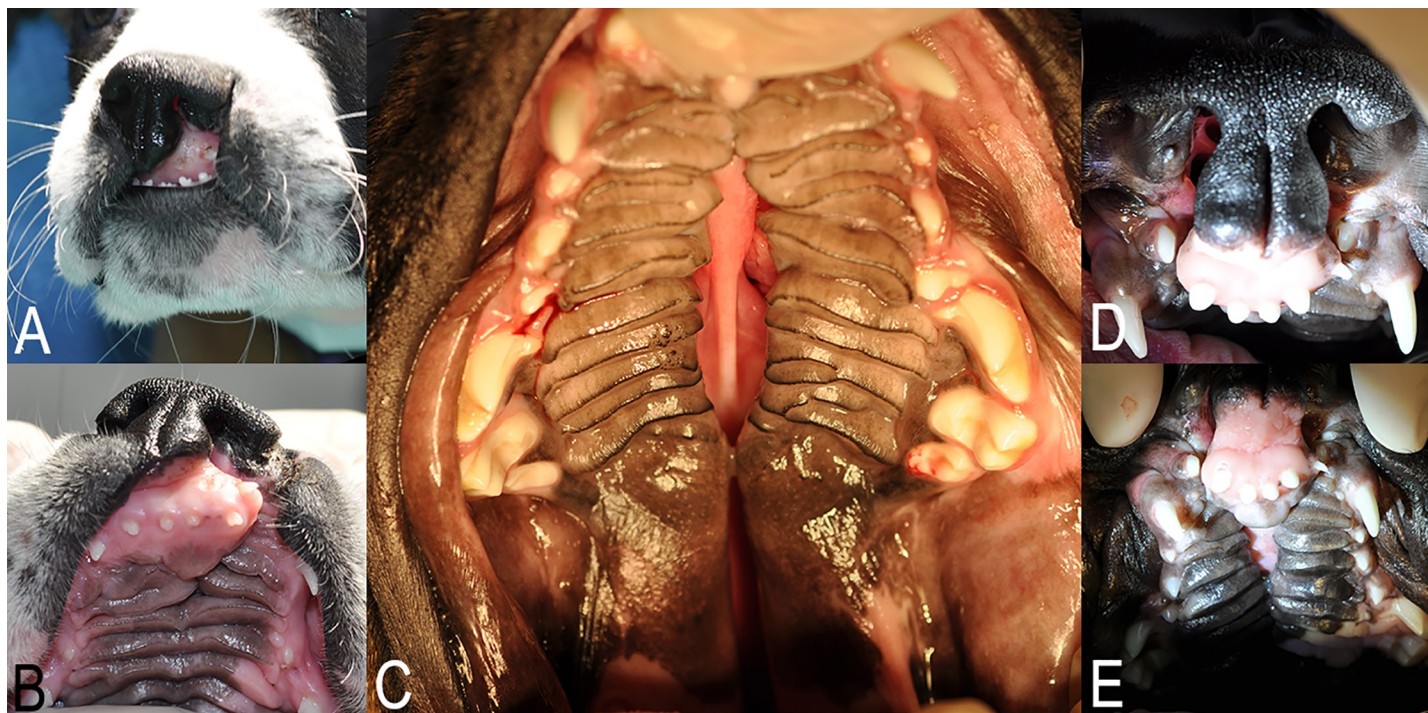

**Fig 1. Representative clinical photographs of orofacial clefts in dogs.** A-B) Extraoral (A) and intraoral (B) appearance of unilateral cleft lip in a dog; C) intraoral appearance of isolated cleft palate in a dog; D-E) extraoral (D) and intraoral (E) appearance of bilateral cleft lip and palate in a dog.

survey was designed to be easy to read, of minimal length, and functional on both computers and mobile devices. Questions were phrased to be concise and unambiguous to make clear what information was being queried. When complex or technical terms could not be replaced with lay language, a clear definition of the term in lay language was provided.

For defining phenotype based on gross cleft morphology, descriptions were supplemented by photographs of the cleft types. Definitions for the different cleft morphologies were provided as follows: CL (lip defects on one or both sides without hard or soft palate defects), CP (defects affecting only the hard and soft palate but not the lips), or CLP (defects affecting the lip on one or both sides as well as the hard and soft palates). Whenever possible, respondents were asked to select from a list of predetermined choices to minimize the variability in response types. However, if a respondent felt that the predetermined selections were inadequate, the option to elaborate on a response in a text field was provided. To avoid social desirability distortion and encourage candid responses, anonymity and confidentiality of responses was assured and respondents could backtrack in the survey [28]. Information about the purpose of the project, the anonymity of responses and contact information for the investigators was provided at the start of the survey, along with an informed consent statement that required respondents to reply in the affirmative to access the survey. For quality control and to assess completion time, a pilot version of the survey was previewed and critiqued by selected breeders prior to use in the study population, with feedback incorporated to revise the survey.

The survey was made available to any respondent with Internet access and the survey address. To maximize responses relevant to the target population of live-born purebred puppies within the United States, e-mail announcements were sent to the U.S.-based parent breed clubs and prominent regional breed clubs for the 100 most popular American Kennel Club (AKC) dog breeds based on the 2016 ranking. The announcement encouraged breed clubs to

publicize the survey to subsidiary clubs and to individual breeding operations. E-mail and telephone calls were used for follow-up communication with participating breed clubs to encourage dissemination and participation. Fliers were also produced and distributed at selected kennel club meetings. The e-mail announcement and fliers directed interested breeders to the online survey instrument. Browser cookies were used to ensure that responses were not duplicates, although breeders of more than one breed could complete the survey for each pertinent breed.

The survey requested data pertaining to live births of purebred puppies whelped in the preceding 12-month period. Information collected through the survey instrument included breed, geographic region (according to U.S. Census Bureau Regions and Divisions), number of litters whelped in the preceding 12 months, total number of live births in those litters, number of litters that included one or more live puppies with an orofacial cleft, sex of affected puppies, presence or absence of other congenital abnormalities in puppies with orofacial clefts, phenotype of the cleft (CL, CP or CLP), whether the defect was diagnosed by a veterinarian or otherwise recognized by the breeder or another person, any perceived cause of the orofacial cleft and estimated financial loss per affected puppy and per year.

## Statistical analysis

Descriptive statistics were calculated, with mean and standard deviation or median, interquartile range, and range reported for continuous variables as appropriate and with categorical variables reported as frequency counts. Breeds were assigned to skull type categories (brachycephalic, mesaticephalic or dolichocephalic) based on described AKC breed standards with reference to other descriptions [29–34]. Breeds were assigned to modern, ancient, herding/sighthound, mountain, or mastiff/terrier genetic clusters based on published phylogenetic analyses [24, 25]. Data were analyzed for univariate associations between cleft type (CL, CP, CLP, and all orofacial clefts) and breed, skull type, and genetic cluster via calculation of odds ratios (ORs) and associated 95% confidence intervals (CIs).

Skull type and genetic cluster associations were secondarily stratified by sex, using an estimated secondary sex ratio of 1:1 in the underlying populations. The $\chi^2$ test of homogeneity was used to determine if a biological interaction between cleft incidence and sex was present (i.e., if the frequency of clefts differed by sex). If no sex effect modification was noted, Mantel-Haenszel adjusted ORs were calculated and were compared to the crude ORs. A difference > 10% between crude and adjusted ORs was taken to indicate confounding by sex. Haldane-Anscombe correction was applied to contingency tables yielding zero cell values in the denominator.

A multivariable logistic regression model was used to assess the joint effects of skull type, genetic cluster and sex on cleft occurrence. The model was first fitted with just skull type and genetic cluster. A model incorporating sex was also assessed but was considered a secondary analysis because the underlying sex distribution was estimated rather than directly queried on the survey instrument. Models were initially constructed with all candidate predictors and all first-order interactions entered; if a significant statistical interaction, defined as $P < 0.05$ for any interaction term, were noted, data were stratified on the offending variable and the model was re-fit to individual strata. For models with no statistical interaction, the interaction terms were dropped, and the model was re-run with only the candidate predictors. Values of $P < 0.05$ were considered statistically significant. Statistical analysis was performed using commercially available software (SAS version 9.4, SAS Institute Inc., Cary, NC).

## Results

There were 974 respondents who completed all or a part of the survey, representing 78 AKC-recognized breeds. Not all respondents answered every question. A total of 22 respondents

indicated that the puppies affected with orofacial clefts also had other congenital abnormalities present, and these data were therefore excluded from further analysis. Of 151 respondents who answered the question about how their animals were diagnosed with an orofacial cleft, clefts were diagnosed by a veterinarian in 104 (68.9%) cases and by the owner in 29 (19.2%) cases. Eighteen (11.9%) respondents selected "other."

Among 7,429 live-born puppies in the 12 preceding months that were included in the study, 228 orofacial clefts (3.0%) were reported, with a phenotypic distribution as follows: CL, 59 (26%), CP 134 (59%), and CLP 35 (15%). The mean ± SD litter size overall was 4.8 ± 2.4. Compared with litter size 2 to 4, litter size 7 to 9 was at decreased odds of CP (OR, 0.39; 95% CI, 0.23 to 0.67), litter size 7 to 9 was at decreased odds of any cleft (OR, 0.45; 95% CI, 0.30 to 0.68), and litter size ≥ 10 was at decreased odds of any cleft (OR, 0.34; 95% CI, 0.12 to 0.95). However, restricting analysis to only those respondents who reported having both litters with and without clefts, there was no statistically significant difference in the litter size between litters with clefts and litters without clefts (Wilcoxon signed rank test $P = 0.2935$). This held true when stratified on genetic cluster ($P = 0.0625$ to 0.75) and skull type ($P = 0.2401$ to 0.7578). The geographic distribution of respondents was summarized (Table 1).

Compared with the United States as a whole, the Midwest region was at decreased odds of any cleft (OR, 0.66; 95% CI, 0.47 to 0.93). No other statistically significant associations were observed between the incidence of orofacial clefts and the geographic location of the breeding program.

The number of live births and orofacial clefts by breed was tabulated (Table 2) for all breeds with at least 60 live births reported. Incidence per 1,000 live births by breed was calculated (Table 3). The Labrador Retriever had the largest number of reported live births and was therefore selected as the reference breed for ORs. Breeds identified as having significantly increased odds of CL compared to the Labrador Retriever were the Boston Terrier (OR, 12.12; 95% CI, 2.33 to 63.01), French Bulldog (OR, 10.67; 95% CI, 2.41 to 47.15), Cavalier King Charles Spaniel (OR, 18.29; 95% CI, 3.65 to 91.65) and English Bulldog (OR, 7.00; 95% CI, 1.27 to 38.48) (Table 4). The Boston Terrier (OR, 4.44; 95% CI, 1.93 to 10.25) and French Bulldog (OR, 4.06; 95% CI, 2.07 to 7.97) also had significantly increased odds of CP compared to the Labrador Retriever. Relatively few cases of CLP were reported, with none in the reference group. Therefore, for any breed with at least one reported case, OR estimates must be considered inflated and imprecise. Breeds having at least one reported CLP were the Boston Terrier, French Bulldog, Papillon, English Bulldog, Australian Shepherd and Italian Greyhound. Compared to Labrador Retrievers, the Boston Terrier, French Bulldog, Cavalier King Charles Spaniel and Papillon had increased odds of any orofacial cleft.

**Table 1. Geographic distribution of survey respondents.**

| Geographic area | No. (%) of respondents |
|---|---|
| **Northeast** (Connecticut, Maine, Massachusetts, New Hampshire, New Jersey, New York, Pennsylvania, Rhode Island, and Vermont) | 158 (16.2) |
| **Midwest** (Iowa, Illinois, Indiana, Kansas, Michigan, Minnesota, Missouri, North Dakota, Nebraska, Ohio, South Dakota, and Wisconsin) | 221 (22.7) |
| **South** (Alabama, Arkansas, District of Columbia, Delaware, Florida, Georgia, Kentucky, Louisiana, Maryland, Mississippi, North Carolina, Oklahoma, South Carolina, Tennessee, Texas, Virginia, and West Virginia) | 251 (25.8) |
| **West** (Alaska, Arizona, California, Colorado, Hawaii, Idaho, Montana, New Mexico, Nevada, Oregon, Utah, Washington, and Wyoming) | 244 (25.1) |
| Other | 100 (10.3) |
| **Total** | **974** |

**Table 2. Number of live births and orofacial clefts by breed.**

| Breed | Live births | CL | CP | CLP | All |
|---|---|---|---|---|---|
| Labrador Retriever | 770 | 2 | 12 | 0 | 14 |
| Boston Terrier | 182 | 5 | 11 | 10 | 26 |
| French Bulldog | 550 | 14 | 32 | 8 | 54 |
| Cavalier King Charles Spaniel | 134 | 6 | 4 | 0 | 10 |
| Papillon | 243 | 3 | 5 | 2 | 10 |
| English Bulldog | 225 | 4 | 4 | 1 | 9 |
| Golden Retriever | 373 | 3 | 8 | 1 | 12 |
| English Cocker Spaniel | 296 | 2 | 5 | 0 | 7 |
| Pembroke Welsh Corgi | 345 | 1 | 7 | 0 | 8 |
| Miniature Schnauzer | 191 | 1 | 3 | 0 | 4 |
| Cardigan Welsh Corgi | 158 | 0 | 3 | 0 | 3 |
| Australian Shepherd | 120 | 0 | 1 | 1 | 2 |
| Shetland Sheepdog | 85 | 0 | 1 | 0 | 1 |
| Dalmatian | 326 | 1 | 2 | 0 | 3 |
| Italian Greyhound | 222 | 0 | 1 | 1 | 2 |
| Weimaraner | 252 | 0 | 2 | 0 | 2 |
| Parson Russell Terrier | 415 | 0 | 2 | 0 | 2 |
| Beagle | 331 | 0 | 1 | 0 | 1 |
| Shiba Inu | 63 | 0 | 0 | 0 | 0 |
| Irish Setter | 87 | 0 | 0 | 0 | 0 |
| Rottweiler | 92 | 0 | 0 | 0 | 0 |
| Brittany | 151 | 0 | 0 | 0 | 0 |
| Poodle | 143 | 0 | 0 | 0 | 0 |
| Doberman Pinscher | 213 | 0 | 0 | 0 | 0 |

CL, cleft lip only; CP, cleft palate only; CLP, cleft lip and palate; All, cleft of any type.

In univariate analysis evaluating genetic cluster (Table 5), breeds in the mastiff/terrier genetic cluster were at increased odds of orofacial clefts of any type (OR, 2.56; 95% CI, 1.85 to 3.53; P < 0.001), CL (OR, 1.77; 95% CI, 1.01 to 3.11; $P$ = 0.0455), CP (OR, 2.62; 95% CI, 1.70 to 4.03; $P$ < 0.001), and CLP (OR, 4.75; 95% CI, 1.81 to 12.46; $P$ < 0.001). Breeds in the ancient breed cluster were at increased odds of orofacial clefts of any type (OR, 2.91; 95% CI, 1.53 to 5.53; P < 0.001) and CP (OR, 4.43; 95% CI, 2.13 to 9.22; $P$ < 0.001). However, within this genetic cluster, data were reported only for a total of 223 livebirths from the following breeds: Afghan hound (n = 4; 1.79%), Akita (17; 7.62%), Basenji (13; 5.83%), Chinese Shar-pei (31; 13.9%), Lhasa Apso (6; 2.69%), Samoyed (73; 32.74%), Shiba Inu (63; 28.25%), and Siberian Husky (16; 7.17%). Within this cluster, all orofacial clefts occurred in just two breeds: the Chinese Shar-pei (n = 3) and Samoyed (9). Univariate associations with skull type were tabulated (Table 6). Brachycephalic breeds were at increased odds of orofacial clefts of any type (OR, 4.75; 95% CI, 3.57 to 6.31; $P$ < 0.001), CL (OR, 9.12; 95% CI, 4.88 to 17.03; $P$ < 0.001), CP (OR, 2.60; 95% CI, 1.81 to 3.73; $P$ < 0.001), and CLP (OR 20.47; 95% CI, 7.18 to 58.31; $P$ < 0.001).

No biological interaction was observed with sex. After stratification for sex, the crude and Mantel-Haenszel adjusted ORs for genetic cluster and skull type were within 10% of each other for all categories of orofacial clefts, indicating that sex was unlikely to be a confounding variable. Multiple logistic regression models omitting sex found no statistically significant interaction between skull type and genetic cluster for any orofacial cleft category. For CL and

**Table 3. Breed incidence and associated 95% confidence intervals of orofacial clefts, per 1,000 live births.**

| Breed | CL | 95% CI | CP | 95% CI | CLP | 95% CI | All | 95% CI |
|---|---|---|---|---|---|---|---|---|
| Australian Shepherd | 0.0 | (0.0, 20.7) | 8.3 | (0.9, 38.3) | 8.3 | (0.9, 38.3) | 16.7 | (3.5, 52.4) |
| Beagle | 0.0 | (0.0, 7.6) | 3.0 | (0.3, 14.0) | 0.0 | (0.0, 7.6) | 3.0 | (0.3, 14.0) |
| Boston Terrier | 27.5 | (9.0, 59.2) | 60.4 | (32.5, 102.2) | 54.9 | (28.6, 95.3) | 142.9 | (97.8, 199.2) |
| Brittany | 0.0 | (0.0, 16.5) | 0.0 | (0.0, 16.5) | 0.0 | (0.0, 16.5) | 0.0 | (0.0, 16.5) |
| Cardigan Welsh Corgi | 0.0 | (0.0, 15.7) | 19.0 | (5.4, 49.8) | 0.0 | (0.0, 15.7) | 19.0 | (5.4, 49.8) |
| Cavalier King Charles Spaniel | 44.8 | (18.9, 90.0) | 29.9 | (10.2, 69.4) | 0.0 | (0.0, 18.5) | 74.6 | (39.0, 128.3) |
| Dalmatian | 3.1 | (0.3, 14.2) | 6.1 | (1.3, 19.5) | 0.0 | (0.0, 7.7) | 9.2 | (2.6, 24.4) |
| Doberman Pinscher | 0.0 | (0.0, 11.7) | 0.0 | (0.0, 11.7) | 0.0 | (0.0, 11.7) | 0.0 | (0.0, 11.7) |
| English Bulldog | 17.8 | (6.0, 41.7) | 17.8 | (6.0, 41.7) | 4.4 | (0.5, 20.6) | 40.0 | (20.0, 71.7) |
| English Cocker Spaniel | 6.8 | (1.4, 21.5) | 16.9 | (6.5, 36.6) | 0.0 | (0.0, 8.4) | 23.6 | (10.6, 45.9) |
| French Bulldog | 25.5 | (14.7, 41.2) | 58.2 | (40.9, 80.1) | 14.5 | (6.9, 27.3) | 98.2 | (75.4, 125.2) |
| Golden Retriever | 8.0 | (2.3, 21.3) | 21.4 | (10.2, 40.1) | 2.7 | (0.3, 12.5) | 32.2 | (17.7, 53.8) |
| Irish Setter | 0.0 | (0.0, 28.4) | 0.0 | (0.0, 28.4) | 0.0 | (0.0, 28.4) | 0.0 | (0.0, 28.4) |
| Italian Greyhound | 0.0 | (0.0, 11.2) | 4.5 | (0.5, 20.9) | 4.5 | (0.5, 20.9) | 9.0 | (1.9, 28.6) |
| Labrador Retriever | 2.6 | (0.5, 8.3) | 15.6 | (8.5, 26.2) | 0.0 | (0.0, 3.3) | 18.2 | (10.5, 29.5) |
| Miniature Schnauzer | 5.2 | (0.6, 24.2) | 15.7 | (4.4, 41.3) | 0.0 | (0.0, 13.0) | 20.9 | (7.1, 49.0) |
| Papillon | 12.3 | (3.5, 32.6) | 20.6 | (7.9, 44.5) | 8.2 | (1.7, 26.1) | 41.2 | (21.4, 71.8) |
| Parson Russell Terrier | 0.0 | (0.0, 6.0) | 4.8 | (1.0, 15.4) | 0.0 | (0.0, 6.0) | 4.8 | (1.0, 15.4) |
| Pembroke Welsh Corgi | 2.9 | (0.3, 13.5) | 20.3 | (9.1, 39.4) | 0.0 | (0.0, 7.2) | 23.2 | (11.0, 43.3) |
| Poodle | 0.0 | (0.0, 17.4) | 0.0 | (0.0, 17.4) | 0.0 | (0.0, 17.4) | 0.0 | (0.0, 17.4) |
| Rottweiler | 0.0 | (0.0, 26.9) | 0.0 | (0.0, 26.9) | 0.0 | (0.0, 26.9) | 0.0 | (0.0, 26.9) |
| Shetland Sheepdog | 0.0 | (0.0, 29.0) | 11.8 | (1.3, 53.7) | 0.0 | (0.0, 29.0) | 11.8 | (1.3, 53.7) |
| Shiba Inu | 0.0 | (0.0, 38.9) | 0.0 | (0.0, 38.9) | 0.0 | (0.0, 38.9) | 0.0 | (0.0, 38.9) |
| Weimaraner | 0.0 | (0.0, 9.9) | 7.9 | (1.7, 25.2) | 0.0 | (0.0, 9.9) | 7.9 | (1.7, 25.2) |

CL, cleft lip only; CI, confidence interval; CP, cleft palate only; CLP, cleft lip and palate; All, cleft of any type.

CLP, skull type ($P < 0.0001$) was found to be a significant predictor, while genetic cluster ($P = 0.365$ for CL; $P = 0.533$ for CLP) was not. For CP and for clefts of any type, both skull type ($P < 0.0001$ for both) and genetic cluster ($P < 0.0001$ for CP; $P = 0.002$ for cleft of any type) were significant predictors.

In the secondary analyses incorporating sex as a predictor, no statistically significant first-order interaction was noted for CL, CP, or CLP (all $P > 0.05$). For CL, in models without interaction, neither genetic cluster ($P = 0.266$) nor sex ($P = 0.402$) were significant predictors, while skull type remained significant ($P < 0.0001$). For CP, skull type ($P < 0.0001$) and genetic cluster ($P = 0.0001$) remained significant predictors, and sex was also a significant predictor ($P = 0.022$), with females having lower odds compared to males. For CLP, neither genetic cluster ($P = 0.525$) nor sex ($P = 0.506$) were significant predictors, while skull type remained significant ($P < 0.0001$).

The multivariable model using any cleft as the outcome and incorporating sex found a statistically significant interaction between genetic cluster and sex. The data were then analyzed separately by genetic cluster. For the modern cluster, both skull type ($P < 0.0001$) and sex ($P = 0.0003$) were significant predictors, with females having lower odds than males (OR, 0.31; 95% CI, 0.16 to 0.58) and with brachycephalic breeds having increased odds relative to mesaticephalic dogs (OR, 6.04, 95% CI, 3.37 to 10.80). In both the ancient cluster and the herding/sighthound cluster, neither skull type ($P = 0.973$ for ancient; $P = 0.861$ for herding/sighthound) nor sex ($P = 0.272$ for ancient; $P = 0.531$ for herding/sighthound) was a significant predictor.

**Table 4. Breed odds ratios and associated 95% confidence intervals for orofacial clefts.**

| Breed | CL OR (95% CI) | CP OR (95% CI) | CLP OR (95% CI) | All OR (95% CI) |
|---|---|---|---|---|
| Labrador Retriever | REF | REF | REF | REF |
| Boston Terrier | 12.12 (2.33–63.01) | 4.44 (1.93–10.25) | 101.77 (43.67–237.15) | 9.00 (4.60–17.63) |
| French Bulldog | 10.67 (2.41–47.15) | 4.06 (2.07–7.97) | 25.91 (10.68–62.84) | 5.88 (3.23–10.70) |
| Cavalier King Charles Spaniel | 18.29 (3.65–91.65) | 2.03 (0.65–6.40) | — | 4.35 (1.89–10.02) |
| Papillon | 4.87 (0.81–29.30) | 1.35 (0.47–3.88) | 16.22 (4.12–63.88) | 2.32 (1.02–5.29) |
| English Bulldog | 7.00 (1.27–38.48) | 1.17 (0.37–3.65) | 10.50 (1.91–57.71) | 2.25 (0.96–5.27) |
| Golden Retriever | 3.14 (0.52–18.88) | 1.40 (0.57–3.45) | 6.28 (1.15–34.46) | 1.80 (0.82–3.92) |
| English Cocker Spaniel | 2.62 (0.37–18.66) | 1.09 (0.38–3.12) | — | 1.31 (0.52–3.27) |
| Pembroke Welsh Corgi | 1.12 (0.10–12.41) | 1.31 (0.51–3.35) | — | 1.28 (0.53–3.08) |
| Miniature Schnauzer | 2.02 (0.18–22.41) | 1.01 (0.28–3.62) | — | 1.16 (0.38–3.55) |
| Cardigan Welsh Corgi | — | 1.22 (0.34–4.37) | — | 1.05 (0.30–3.68) |
| Australian Shepherd | — | 0.53 (0.07–4.14) | 19.22 (3.48–106.11) | 0.92 (0.21–4.08) |
| Shetland Sheepdog | — | 0.75 (0.10–5.84) | — | 0.64 (0.08–4.95) |
| Dalmatian | 1.17 (0.11–12.95) | 0.39 (0.09–1.75) | — | 0.50 (0.14–1.76) |
| Italian Greyhound | — | 0.29 (0.04–2.21) | 10.31 (1.88–56.66) | 0.49 (0.11–2.18) |
| Weimaraner | — | 0.50 (0.11–2.27) | — | 0.43 (0.10–1.91) |
| Parson Russell Terrier | — | 0.31 (0.07–1.37) | — | 0.26 (0.06–1.16) |
| Beagle | — | 0.19 (0.02–1.47) | — | 0.16 (0.02–1.25) |

CL, cleft lip only; CI, confidence interval; CP, cleft palate only; CLP, cleft lip and palate; All, cleft of any type; OR, odds ratio; REF, reference group. A dash (—) indicates no live births of the given orofacial cleft.

For the mastiff/terrier group, sex ($P = 0.464$) was not a significant predictor, whereas skull type ($P < 0.0001$) was, with brachycephalic dogs having higher odds than mesaticephalic dogs (OR, 4.99; 95% CI, 3.31 to 7.52).

**Table 5. Genetic cluster odds ratios and associated 95% confidence intervals for orofacial clefts.**

| Cluster | Live births | No. of CL | CL OR (95% CI) | No. of CP | CP OR (95% CI) | No. of CLP | CLP OR (95% CI) | Total clefts | Overall OR (95% CI) |
|---|---|---|---|---|---|---|---|---|---|
| Ancient | 223 | 1 | 0.65 (0.09–4.89) | 10 | 4.43 (2.13–9.21)* | 1 | 2.49 (0.29–21.38) | 12 | 2.91 (1.53–5.53)* |
| M | | 0 | 0 (0–0) | 4 | 3.38 (1.17–9.79) | 1 | 4.95 (0.57–42.73) | 5 | 2.40 (0.94–6.13) |
| F | | 1 | 1.31 (0.17–9.86) | 6 | 5.12 (2.08–12.59) | 0 | 0 (0–0) | 7 | 3.43 (1.52–7.72) |
| Herd/Sight | 1169 | 1 | 0.12 (0.02–0.93) | 18 | 1.48 (0.82–2.67) | 2 | 0.95 (0.18–4.88) | 21 | 0.94 (0.56–1.56) |
| M | | 0 | 0 (0–0) | 11 | 1.78 (0.88–3.57) | 1 | 0.94 (0.11–8.10) | 12 | 0.80 (0.39–1.63) |
| F | | 1 | 0.25 (0.03–1.85) | 7 | 1.13 (0.49–2.60) | 1 | 0.95 (0.11–8.11) | 9 | 1.07 (0.57–2.02) |
| Mastiff/Terrier | 2815 | 34 | 1.77 (1.01–3.11)* | 76 | 2.62 (1.70–4.03)* | 24 | 4.75 (1.81–12.46)* | 134 | 2.56 (1.85–3.53)* |
| M | | 17 | 1.82 (0.92–3.41) | 41 | 2.78 (1.72–4.50) | 13 | 5.16 (1.84–14.51) | 71 | 2.72 (1.89–3.90) |
| F | | 17 | 1.77 (0.92–3.41) | 35 | 2.37 (1.45–3.90) | 11 | 4.37 (1.51–12.59) | 63 | 2.40 (1.65–3.48) |
| Modern | 2765 | 19 | REF | 29 | REF | 5 | REF | 53 | REF |
| M | | 14 | | 24 | | 4 | | 42 | |
| F | | 5 | | 5 | | 1 | | 11 | |
| Other | 457 | 3 | 0.96 (0.28–3.24) | 0 | 0 (0–0) | 3 | 3.65 (0.87–15.32) | 6 | 0.68 (0.29–1.59) |
| M | | 2 | 1.28 (0.30–5.51) | 0 | 0 (0–0) | 2 | 4.87 (0.94–25.26) | 4 | 0.91 (0.33–2.54) |
| F | | 1 | 0.64 (0.08–4.77) | 0 | 0 (0–0) | 1 | 2.43 (0.28–20.86) | 2 | 0.45 (0.11–1.87) |

CL, cleft lip only; CI, confidence interval; CP, cleft palate only; CLP, cleft lip and palate; F, female; M, male; OR, odds ratio; REF, reference group. Asterisks (*) indicate significantly increased odds.

**Table 6. Skull type odds ratios and associated 95% confidence intervals for orofacial clefts.**

| Skull type | Live births | No. of CL | OR (95% CI) | No. of CP | OR (95% CI) | No. of CLP | OR (95% CI) | Total clefts | OR (95% CI) |
|---|---|---|---|---|---|---|---|---|---|
| Mesaticephalic | 4459 | 13 | REF | 63 | REF | 4 | REF | 80 | REF |
| M | | 9 | | 38 | | 2 | | 49 | |
| F | | 4 | | 25 | | 2 | | 31 | |
| Brachycephalic | 1617 | 42 | 9.12 (4.88–17.03)* | 58 | 2.60 (1.81–3.73) | 29 | 20.47 (7.18–58.31) | 129 | 4.75 (3.57–6.31) |
| M | | 22 | 6.90 (3.16–15.05) | 35 | 2.61 (1.64–4.16) | 17 | 23.92 (5.51–103.77) | 74 | 4.48 (3.10–6.49) |
| F | | 20 | 14.11 (4.81–41.42) | 23 | 2.58 (1.46–4.58) | 12 | 16.78 (3.75–75.14) | 55 | 5.18 (3.31–8.10) |
| Dolichocephalic | 1042 | 2 | 0.66 (0.15–2.92) | 11 | 0.75 (0.39–1.42) | 1 | 1.07 (0.12–9.57) | 14 | 0.75 (0.42–1.32) |
| M | | 2 | 0.95 (0.20–4.41) | 6 | 0.67 (0.28–1.60) | 1 | 2.14 (0.19–23.67) | 9 | 0.78 (0.38–1.60) |
| F | | 0 | 0 (0–0) | 5 | 0.85 (0.33–2.24) | 0 | 0 (0–0) | 5 | 0.69 (0.27–1.78) |
| Unspecified | 311 | 1 | 1.10 (0.14–8.46) | 1 | 0.23 (0.03–1.65) | 1 | 3.59 (0.40–32.18) | 3 | 0.53 (0.17–1.70) |
| M | | 0 | 0 (0–0) | 0 | 0 (0–0) | 0 | 0 (0–0) | 0 | 0 (0–0) |
| F | | 1 | 3.60 (0.40–32.42) | 1 | 0.57 (0.08–4.24) | 1 | 7.21 (0.65–79.94) | 3 | 1.40 (0.42–4.62) |

CL, cleft lip only; CI, confidence interval; CP, cleft palate only; CLP, cleft lip and palate; F, female; M, male; OR, odds ratio; REF, reference group.

Asterisks (*) indicate significantly increased odds.

The median financial loss represented by one affected animal was $2,000 (mean, $2,249; IQR, $1,000; range, $0 to $10,000). The median financial loss per respondent (limiting analysis to those reporting at least one affected animal) was $2,500 (mean, $3,725; IQR, $2,500; range, $0 to $21,000).

## Discussion

The present study constitutes the largest epidemiological investigation of orofacial clefts in purebred dogs available to date in the scientific literature. These results provide new insights regarding these relatively common and clinically relevant congenital malformations and thus may be useful for directing future studies, such as those aimed at further documenting their epidemiological, economic, or comparative features; at elucidating the embryological or genetic mechanisms involved; or at proposing novel therapeutic or preventive strategies.

Importantly, the results of this study show that the overall incidence of orofacial clefts in dogs varies significantly across breeds, ranging from none to a few cases per 1,000 live births in some, to several dozen cases per 1,000 live births in others. This finding conflicts with the notion that the incidence of orofacial clefts in dogs is comparable to that reported in humans [5, 21]. For example, a frequently cited study found an incidence of 1.1 cases per 1,000 live births in dogs [35], similar to the 1.7 cases per 1,000 live births observed in humans [36]. However, this information derives from a colony of Beagles that were bred with the intent of perpetuating a CLP phenotype for research applications and therefore does not represent the population of dogs bred for other purposes and may not be generalizable to other breeds.

A relatively high frequency of orofacial clefts has been reported in certain breeds, including Boston Terriers, Pyrenees Shepherd Dogs, and Boxers [4, 22, 26]. However, these and similar reports are typically limited to individual dog families or breeding programs that likely include common ancestors among the progeny and thus cannot be used to infer incidence within specific breeds. Conversely, given that the present study captured nationwide participation from multiple independent breeding programs representing numerous breeds, results of the study reported here offer a less biased estimation of the actual incidence within represented breeds and constitute credible evidence of significant variability across breeds.

The overall frequency of orofacial clefts observed in this study varied according to phenotype (i.e., CL vs. CP vs. CLP), which is consistent with a previous observation and reinforces the notion that the CP phenotype is more common in dogs compared to CL and CLP [12]. However, results also showed that this pattern may only apply to certain breeds (e.g., Labrador Retriever, Pembroke Welsh Corgi, and French Bulldog) but not others (e.g., Boston Terrier, Cavalier King Charles Spaniel, English Bulldog). Given that the CP phenotype is believed to be genetically distinct from CL and CLP [36, 37], these results suggest that orofacial clefts are phenotypically and genotypically heterogeneous among breeds. From a comparative perspective, these results contrast with the cleft phenotype distribution reported in humans, in which CLP is the most common form, representing approximately 43% of the cases, while CL and CP represent 26% and 31%, respectively [38]. This would suggest differences in the genetic origin of orofacial clefts between humans and at least some breeds of dogs. This fits with the evolutionary theory for the different dog breeds.

Based both on univariate and multivariate analyses, skull type was strongly associated with the incidence of orofacial clefts in this study. Specifically, the odds of CP, CL and CLP were consistently and significantly higher in the brachycephalic group compared to the reference skull type group (i.e., mesaticephalic). This is a relevant finding because it confirms numerous anecdotal reports that have suggested that brachycephalic dogs are predisposed to orofacial clefts [2, 9]. However, these findings should be contextualized for a more accurate interpretation. First, it should be noted that there is no universally accepted definition of what constitutes a brachycephalic, mesaticephalic, or dolichocephalic dog [34]. To minimize this potential limitation, reference was made to published studies in which skull type assignment had been done using reproducible measures [12, 29, 32–34]. Breeds for which no reference data were found were left unassigned (i.e. unspecified). Second, the incidence of orofacial clefts in some mesaticephalic breeds should not be overshadowed by the relatively high incidence in brachycephalic breeds. That is, orofacial clefts were not uncommon in at least some of the mesaticephalic breeds represented in this study (Labrador Retriever, Cardigan Welsh Corgi, Miniature Schnauzer, etc.). Conversely, dolichocephalic breeds had the lowest incidence of orofacial clefts when compared with the other two skull types. This finding is consistent with a previous observation suggesting that orofacial clefts are uncommon in dolichocephalic dogs [12].

Another interesting finding of this study was the strong association of the mastiff/terrier genetic cluster with all of the orofacial cleft phenotypes. This finding was not surprising considering that, with few exceptions, the breeds within this genetic cluster have been reported to be predisposed to orofacial clefts, including Labrador Retrievers, Yorkshire Terriers, Mastiffs, Bullmastiffs, Staffordshire Bull Terriers, Bull Terriers, West Highland White Terriers, Scottish Terriers, and Golden Retrievers, among several others [2, 25]. Moreover, the mastiff/terrier genetic cluster contains many of the most popular brachycephalic breeds, including all of those found to be associated with a high incidence of orofacial clefts in this study (i.e., Boston Terrier, English Bulldog, French Bulldog) [25]. These findings raise the question of whether brachycephalic dogs belonging to other genetic clusters (e.g., Pug, Pekingese) are equally predisposed to orofacial clefts as those within the mastiff/terrier cluster.

Compared with other genetic clusters, breeds within the herding/sighthound genetic cluster had the lowest incidence of orofacial clefts. This is likely explained by the fact that many of the breeds in this group are dolichocephalic [25], which appear not to be predisposed to orofacial clefts [12]. On the other hand, the association observed between orofacial clefts and the ancient genetic cluster is intriguing, considering that all dog breeds, including those without apparent genetic predisposition to orofacial clefts, derive from this lineage. This could suggest that risk or causative alleles appeared within certain specific ancient breeds (e.g. Samoyed and Chines Shar-pei) after more recent clades had branched off the ancestral root. However, the sample

size of the ancient genetic cluster in this study was by far the lowest. Additional studies will be required to validate these observations, especially considering that the results might have been biased by a potentially outlier sample.

Regarding a possible association between sex and incidence of orofacial clefts in dogs, the univariate and multivariate analyses showed somewhat different results. Univariate analyses showed that the incidence was similar in males and females regardless of defect phenotype. This finding would be consistent with previous observations [4, 12], but contrasts with known variations in humans in which a) orofacial clefts in general are more common in males (58%) compared to females (42%); and b) CL and CLP are more common in males compared to females, and CP is more in females compared to males [38–40]. These findings would again suggest that the potential genetic bases of orofacial clefts differ in humans and dogs. Alternatively, multivariate analyses showed that females had lower odds than males within the modern genetic cluster as well as in cases of CP. The latter finding is the opposite of what is observed in humans and thus would also suggest genetic differences between both species.

In terms of geographic location, it was interesting to observe that the Midwest region was at decreased odds of any orofacial clefts. Although the reasons for this cannot be ascertained based on this study, possible factors include reduced exposure to geographic-dependent teratogenic or infectious agents, differences in husbandry practices or nutrition, or genetically distinct breeding lines with a lower frequency of risk or causative alleles. It is also possible that certain breeds are better represented in particular geographic parts of the United States (i.e., breed and geographic location are not independent). Additional studies will help elucidate how the geographic location of breeding programs might be associated with the incidence of orofacial clefts in purebred dogs.

Even though the financial burden of orofacial clefts in breeding operations cannot be determined based solely on the information collected in this study, the individual losses reported by breeders were noteworthy. Moreover, orofacial clefts were significantly more common in relatively small litters typical of brachycephalic breeds, which likely amplifies the financial impact represented by every affected individual.

The present study focused on purebred dogs bred in the United States over a period of 12 months. Therefore, additional studies will be required to determine the incidence patterns of orofacial clefts in purebred dogs in other parts of the world, of non-purebred dogs both inside and outside of the United States, and over longer periods of time. Indeed, significant epidemiological variability based on geographic location and ethnicity [38, 40] as well as yearly fluctuations [39] are well documented in humans, and it seems reasonable to assume that dogs are subject to similar factors. Additionally, some of the incidence data shown may have been influenced by response bias. That is, some breeders may have been predisposed to respond in a certain way, or to participate or not, based on whether they had observed a low or a high incidence of orofacial clefts within their breeding operations. The direction of bias is difficult to anticipate: some breeders with a high incidence of clefts might be motivated to participate out of a desire to improve the breed, whereas some might be reluctant to participate to avoid negative publicity for the breed. Similarly, the magnitude of bias is difficult to determine, but is anticipated to be small given the survey instrument design and implementation (guarantee of anonymity, ability to backtrack, widespread dissemination with large numbers of respondents), consistency of responses across breeders within most breeds, and concurrence of the observed data with previous reports and clinical perceptions. Therefore, the incidence results reported here should be used as an estimation that can be used as a basis when designing larger-scale epidemiological studies aimed at establishing more precise incidence patterns within specific breeds of interest.

Overall, the results of this study indicate that breed, genetic cluster, and skull type are of importance in the development of orofacial clefts in dogs. Breeds in the mastiff/terrier genetic

cluster and brachycephalic breeds are predisposed to orofacial clefts, certain breeds in the ancient genetic cluster may be at increased odds of orofacial clefts, male purebred dogs may be predisposed to CPs, and orofacial clefts represent a considerable financial loss to breeders.

## Supporting information

**S1 File. Questionnaire.pdf.** Copy of the online survey used to collect the data reported in this study.
(PDF)

## Acknowledgments

The authors would like to thank participating breed clubs and associations for assistance in distribution of the survey.

## Author Contributions

**Conceptualization:** Nicholas Roman, Patrick C. Carney, Nadine Fiani, Santiago Peralta.

**Data curation:** Nicholas Roman, Patrick C. Carney, Santiago Peralta.

**Formal analysis:** Nicholas Roman, Patrick C. Carney, Santiago Peralta.

**Investigation:** Patrick C. Carney, Santiago Peralta.

**Methodology:** Nicholas Roman, Patrick C. Carney, Nadine Fiani, Santiago Peralta.

**Project administration:** Patrick C. Carney, Santiago Peralta.

**Supervision:** Patrick C. Carney, Santiago Peralta.

**Validation:** Patrick C. Carney, Santiago Peralta.

**Writing – original draft:** Nicholas Roman, Patrick C. Carney, Santiago Peralta.

**Writing – review & editing:** Nicholas Roman, Patrick C. Carney, Nadine Fiani, Santiago Peralta.

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
