## [Decision Letter · Decision Letter 0]

7 Oct 2019

PONE-D-19-22090

Incidence patterns of orofacial clefts in purebred dogs

PLOS ONE

Dear Dr Peralta

Thank you for submitting your manuscript to PLOS ONE. After careful consideration, we feel that it has merit but does not fully meet PLOS ONE’s publication criteria as it currently stands. Therefore, we invite you to submit a revised version of the manuscript that addresses the points raised during the review process.

Many thanks for submitting your manuscript to PLOS One

It was reviewed by two experts in the field, and they have both suggested very minor comments, which is testament to the excellent manuscript submitted

If you could make these minor revisions, if you see fit, then resubmit it, I can recommend it for accepting without the need for re-review.

If you are just making the changes suggested, please just have a single line in the review comments saying so, rather than write a full rebuttal

I look forward to seeing the revisions

Many thanks

Simon

We would appreciate receiving your revised manuscript by Nov 21 2019 11:59PM. To enhance the reproducibility of your results, we recommend that if applicable you deposit your laboratory protocols in protocols.io, where a protocol can be assigned its own identifier (DOI) such that it can be cited independently in the future. For instructions see: http://journals.plos.org/plosone/s/submission-guidelines#loc-laboratory-protocols

A marked-up copy of your manuscript that highlights changes made to the original version. This file should be uploaded as separate file and labeled 'Revised Manuscript with Track Changes'.An unmarked version of your revised paper without tracked changes. This file should be uploaded as separate file and labeled 'Manuscript'.

We look forward to receiving your revised manuscript.

Kind regards,

Simon Russell Clegg, PhD

Academic Editor

PLOS ONE

Journal Requirements:

Reviewers' comments:

Reviewer's Responses to Questions

**Comments to the Author**

1. Is the manuscript technically sound, and do the data support the conclusions?

Reviewer #1: Yes

Reviewer #2: Yes

2. Has the statistical analysis been performed appropriately and rigorously? 

Reviewer #1: Yes

Reviewer #2: Yes

3. Have the authors made all data underlying the findings in their manuscript fully available?

Reviewer #1: Yes

Reviewer #2: Yes

4. Is the manuscript presented in an intelligible fashion and written in standard English?

Reviewer #1: Yes

Reviewer #2: Yes

5. Review Comments to the Author

Reviewer #1: This article is the first broad epidemiological study of orofacial clefts in dogs, providing insights for future investigations regarding different aspects of this congenital abnormality.

The only changes required are as follows:

Lines 61 and 62 (caption): C) Intraoral view ...; D-E) Extraoral and intraoral views …

Reviewer #2: This is a very interesting paper, which is well analysed and well written throughout. I have made a few very minor comments, but if the editor is happy, I am more than happy just to let him have a look at it when resubmitted and then he can accept it without me re-reviewing it. The comments are just minor grammatical things (due to the excellent nature of the manuscript). However, if you disagree please do not feel pressured to change them

Well done on producing a very good, interesting manuscript

Line 20 and 21- a comma maybe better than a semi colon

Line 29-30, a set of brackets may be better than hyphens

Line 41- you say useful models of diseases- maybe expand on which diseases?

Line 49- a comma, or brackets may be better than a hyphen

Line 86- a comma may work better than a semi-colon

Line 111- if a question queried information that could …etc- I am not quite sure that I follow this sentence so could it be reworded somehow?

Line 115- of instead of or maybe?

Line 121- no capital needed for internet

Line 156- maybe to determine if a biological interaction between ….

Line 167- if a significant statistical interaction …..

Line 191- maybe a new sentence instead of a semi colon?

Line 202- calculated may be better than tabulated for the incidence?

Line 205- I wondered if having the values in for the OR for the different breeds may help- but may make it less clear? I can leave that to your judgement

Lines 207-209- the line starting relatively few cases of …. Is a little unclear- maybe consider revising it?

Throughout- should breed names be capitalised?

Line 245- maybe a comma after interaction

Line 252- …incorporating sex found a statistically significant …..

Paragraph 277-285- you talk about rations in other studies, could you put an accurate one from your study in here?

Line 292- I am not a fan of using our, us, we etc in scientific writing

Line 306- does this fit with the evolutionary theory for the different dog breeds?

Line 341- which appear not to be (may sound better)

Line 342- on the other hand sounds a bit colloquial, maybe alternatively?

Paragraph 363- Is the geographic issue associated with breeds being associated with certain parts of the USA?

6. PLOS authors have the option to publish the peer review history of their article (what does this mean?). If published, this will include your full peer review and any attached files.

Reviewer #1: Yes: Prof. Enio Moura

Service of Medical Genetics

Course of Veterinary Medicine

School of Life Sciences

Pontifícia Universidade Católica do Paraná (PUCPR)

Curitiba, PR, Brazil

Reviewer #2: No

---

## [Author Response · Author response to Decision Letter 0]

14 Oct 2019

Reviewer #1: This article is the first broad epidemiological study of orofacial clefts in dogs, providing insights for future investigations regarding different aspects of this congenital abnormality.

The only changes required are as follows:

Lines 61 and 62 (caption): C) Intraoral view ...; D-E) Extraoral and intraoral views …

Response: Thank you very much for reviewing our work and noticing the error in the Figure 1 legend. We have revised the it accordingly.

Reviewer #2: This is a very interesting paper, which is well analysed and well written throughout. I have made a few very minor comments, but if the editor is happy, I am more than happy just to let him have a look at it when resubmitted and then he can accept it without me re-reviewing it. The comments are just minor grammatical things (due to the excellent nature of the manuscript). However, if you disagree please do not feel pressured to change them

Well done on producing a very good, interesting manuscript

Response: Many thanks for the thoughtful comments and meticulous review of our work.

Line 20 and 21- a comma maybe better than a semi colon

Response: Change made as suggested.

Line 29-30, a set of brackets may be better than hyphens

Response: Change made as suggested.

Line 41- you say useful models of diseases- maybe expand on which diseases?

Response: Change made as suggested.

Line 49- a comma, or brackets may be better than a hyphen

Response: Change made as suggested.

Line 86- a comma may work better than a semi-colon

Response: Change made as suggested.

Line 111- if a question queried information that could …etc- I am not quite sure that I follow this sentence so could it be reworded somehow?

Response: Change made as suggested.

Line 115- of instead of or maybe?

Response: Change made as suggested.

Line 121- no capital needed for internet

Response: We kept “Internet” based on the Merriam-Webster dictionary.

Line 156- maybe to determine if a biological interaction between ….

Response: Change made as suggested.

Line 167- if a significant statistical interaction …..

Response: Change made as suggested.

Line 191- maybe a new sentence instead of a semi colon?

Response: Change made as suggested.

Line 202- calculated may be better than tabulated for the incidence?

Response: Change made as suggested.

Line 205- I wondered if having the values in for the OR for the different breeds may help- but may make it less clear? I can leave that to your judgement

Response: Change made as suggested.

Lines 207-209- the line starting relatively few cases of …. Is a little unclear- maybe consider revising it?

Response: Change made as suggested.

Throughout- should breed names be capitalised?

Response: Change made as suggested.

Line 245- maybe a comma after interaction

Response: Change made as suggested.

Line 252- …incorporating sex found a statistically significant …..

Response: Change made as suggested.

Paragraph 277-285- you talk about rations in other studies, could you put an accurate one from your study in here?

Response: Change made as suggested.

Line 292- I am not a fan of using our, us, we etc in scientific writing

Response: Change made as suggested.

Line 306- does this fit with the evolutionary theory for the different dog breeds?

Response: Change made as suggested.

Line 341- which appear not to be (may sound better)

Response: Change made as suggested.

Line 342- on the other hand sounds a bit colloquial, maybe alternatively?

Response: Change made as suggested.

Paragraph 363- Is the geographic issue associated with breeds being associated with certain parts of the USA?

Response: Change made as suggested.

---

## [Editor Report · Decision Letter 1]

17 Oct 2019

Incidence patterns of orofacial clefts in purebred dogs

PONE-D-19-22090R1

Dear Dr. Peralta

We are pleased to inform you that your manuscript has been judged scientifically suitable for publication and will be formally accepted for publication once it complies with all outstanding technical requirements.

With kind regards,

Simon Russell Clegg, PhD

Academic Editor

PLOS ONE

Additional Editor Comments (optional):

Many thanks for resubmitting your manuscript to PLOS One, and for the detailed response to reviewers

As you have responded to the comments, I have recommended your article be accepted for publication

I wish to thank you for your efforts and for submitting a very interesting manuscript

I wish you all the best for your future research and will keep an eye out for future papers from your group

With very best wishes

Simon
---

## [Editor Report · Acceptance letter]

23 Oct 2019

PONE-D-19-22090R1 

Incidence patterns of orofacial clefts in purebred dogs 

Dear Dr. Peralta:

I am pleased to inform you that your manuscript has been deemed suitable for publication in PLOS ONE. Congratulations! Your manuscript is now with our production department. 

With kind regards,

on behalf of

Dr. Simon Russell Clegg 

Academic Editor

PLOS ONE